# Long-Term Tri-Modal In Vivo Tracking of Engrafted Cartilage-Derived Stem/Progenitor Cells Based on Upconversion Nanoparticles

**DOI:** 10.3390/biom11070958

**Published:** 2021-06-29

**Authors:** Chu-Hsin Chen, Na Tang, Ke Xue, Hui-Zhong Zhang, Ya-Hong Chen, Peng Xu, Kang Sun, Ke Tao, Kai Liu

**Affiliations:** 1Department of Plastic and Reconstructive Surgery, Shanghai Ninth People’s Hospital, Shanghai Jiao Tong University School of Medicine, Shanghai 200011, China; chenchuhsin@sjtu.edu.cn (C.-H.C.); 118070@sh9hospital.org.cn (K.X.); 806544782@alumni.sjtu.edu.cn (H.-Z.Z.); 819108@sh9hospital.org.cn (Y.-H.C.); xupeng3241135@sjtu.edu.cn (P.X.); 2State Key Lab of Metal Matrix Composites, Shanghai Jiao Tong University, Shanghai 200240, China; natang@sjtu.edu.cn (N.T.); ksun@sjtu.edu.cn (K.S.)

**Keywords:** cartilage-derived stem/progenitor cells, stem cell tracking system, upconversion nanoparticles, upconversion luminescence, magnetic resonance imaging, computed tomography

## Abstract

Cartilage-derived stem/progenitor cells (CSPCs) are a potential choice for seed cells in osteal and chondral regeneration, and the outcomes of their survival and position distribution in vivo form the basis for the investigation of their mechanism. However, the current use of in vivo stem cell tracing techniques in laboratories is relatively limited, owing to their high operating costs and cytotoxicity. Herein, we performed tri-modal in vivo imaging of CSPCs during subcutaneous chondrogenesis using upconversion nanoparticles (UCNPs) for 28 days. Distinctive signals at accurate positions were acquired without signal noise from X-ray computed tomography, magnetic resonance imaging, and upconversion luminescence. The measured intensities were all significantly proportional to the cell numbers, thereby enabling real-time in vivo quantification of the implanted cells. However, limitations of the detectable range of cell numbers were also observed, owing to the imaging shortcomings of UCNPs, which requires further improvement of the nanoparticles. Our study explores the application value of upconversion nanomaterials in the tri-modal monitoring of implanted stem cells and provides new perspectives for future clinical translation.

## 1. Introduction

Recently, stem cells have been extensively studied as potential therapeutics for multiple osteo and chondral diseases, including metabolic disorders such as degenerative osteoarthrosis and even vitamin D-deficiency rickets [1,2]. Among stem cells, cartilage-derived stem/progenitor cells (CSPCs) tend to be a more ideal choice for seeding cells than the conventionally used bone marrow-derived stem cells (BMSCs). BMSCs typically result in vascularization and ossification, which are adverse for chondrogenesis. In contrast, CSPCs lack the expression of hypertrophic markers, such as RUNX2 and collagen X, thereby preventing terminal differentiation of CSPCs [3,4].

In addition to examining the therapeutic effects of the specimens harvested from animals, real-time tracking of the implanted cells is also necessary to disclose the contribution of the implanted cells toward tissue regeneration and construction. Namely, the survival, viability, position, and quantification of the engrafted cells are all key evidence of the correlation between the observed effects and the transplanted cells [5,6].

Molecular imaging, which is mainly composed of labeling probes and imaging devices, has made it possible to detect living cells spatially, temporally, and especially, noninvasively [7]. However, there are various limitations regarding well-known probes, such as unstable signals and the cytotoxicity of fluorescent proteins and luciferase, the radiation damage and half-life of radionuclides, and the strong cytotoxicity of gold nanoparticles and quantum dots [8].

In the last decade, upconversion nanoparticles (UCNPs) have attracted increased attention as molecular imaging probes for their unique optical properties and biocompatibility [9]. UCNPs are compounds that primarily comprise lanthanide elements from rare earth materials, and they can emit visible and near-infrared (NIR) (650–900 nm) light when evoked by light with a wavelength of 980 nm, which is known as upconversion luminescence (UCL). UCL has a significantly higher photostability, sensitivity, and tissue permeability, and lower autofluorescence and photodamage than the traditional “downconversion” fluorophores, such as GFP and luciferase [10]; additionally, UCNPs themselves have a considerably lower cytotoxicity. More importantly, owing to the paramagnetic capability of gadolinium and high X-ray absorption coefficients of ytterbium, UCNPs doped with Gd and Yb can work as both X-ray computed tomography (CT) and magnetic resonance imaging (MRI) contrast agents, thereby enabling tri-modal in vivo imaging [11,12,13,14].

Although UCNPs possess the unique potential for tri-modal tracking, more focus has been placed on photodynamic therapy [15,16] instead of stem cell tracking or CSPC tracking. In most studies on multimodal in vivo imaging using UCNP, plain nanoparticles are typically administered through intravenous injection for biodistribution assay or tumor-targeted imaging [12,15,16,17,18]; alternatively, they are injected subcutaneously for short-term monitoring [13,19,20,21] or lymphatic imaging [22,23]. Few studies have been conducted on UCNP-labeled stem cells. Hu et al. observed UCNP-labeled BMSCs on implant specimens harvested 14 days after subcutaneous chondrogenesis [24]. Wang et al. conducted lung-targeted therapy and cell imaging using UCNP-labeled human amniotic fluid stem cells through intravenous injection [25]. In the literature regarding cell tracking using MRI and CT, little has been reported on the intensity measurement of labeled cells [25,26]. Thus, as shown in Figure 1, the aim of this study was to investigate the effects of UCNPs on CSPCs, and to explore the feasibility of tracking engrafted CSPCs in a classic ectopic chondrogenesis method by applying UCL, CT, and MRI in vivo.

## 2. Materials and Methods

### 2.1. Fabrication of UCNPs

#### 2.1.1. Synthesis of NaYF_4_:Yb^3+^/Tm^3+^@ NaGdF_4_ @ SiO_2_ UCNPs

NaYF_4_:25%Yb, 0.3% Tm was prepared according to a previously developed procedure [11]. This method was modified to prepare the core/shell structure NaYF_4_:25%Yb, 0.3%Tm/NaGdF_4_ nanocrystals. Typically, both the core precursor solution, which included CF_3_COONa (1 mmol), (CF_3_COO)_3_Y (0.747 mmol), (CF_3_COO)_3_Yb (0.25 mmol), (CF_3_COO)_3_Tm (0.003 mmol), 1-octadecene (2.5 mL), and oleic acid (2.5 mL), and the shell precursor solution, which included (CF_3_COO)_3_Gd (1 mmol), sodium oleate (0.63 mmol), 1-octadecene (7.5 mL), and oleic acid (7.3 mL), were prepared by stirring at 100 °C under vacuum conditions for approximately 1 h until the mixtures became clear. The reaction solution, which included sodium oleate (0.63 mmol), 1-octadecene (10 mL), and oleic acid (9.8 mL), was heated to 290 °C while being stirred under dry nitrogen before the core precursor was injected into it. After the injection, the solution was heated to 330 °C quickly and reacted for 10 min before the injection of the shell precursor. The solution was allowed to react for another 2 min at the end of the injection. The final UCNP solutions were cooled to room temperature before being washed with chloroform and anhydrous ethanol three times. The collected UCNPs were dispersed in cyclohexane for further use. 

#### 2.1.2. Surface Modification of UCNP@SiO_2_

First, 0.5 mL of Igepal Co520 was added to 18 mL of UCNP cyclohexane solution and stirred for 30 min. This was followed by the injection of 20 µL of 3-aminopropyltriethoxysilane (APTES) and ammonia, and a thin silica (SiO_2_) layer was then coated onto the UCNPs. The silica-layer-coated UCNPs are referred to as UCNPs@SiO_2_, and they have a pH value of 8–9. The emulsion was stirred at room temperature for 48 h, and the nanoparticles were then collected by centrifugation (8500 rpm, 6 min) and washed with water three times before being freeze-dried to powder for further use. Before incubation with CSPCs, the UCNPs@SiO_2_ powder was dispersed in a serum-free culture medium (2 mg/mL) via sonication. DNA Transfectin 3000 (TS3000, Herogen Biotechnology Inc. Shanghai, China) was added at a volume ratio of 3:500. The mixture was cultured for 15 min at room temperature before use, and the resultant nanoparticles are denoted as UCNP@SiO_2_-TS.

#### 2.1.3. Characterization of UCNP@SiO_2_-TS

The size and morphology of the nanoparticles were observed using a transmission electron microscope (TEM, JEM-2100, JEOL, Tokyo, Japan) operated at 200 kV. Upconversion fluorescence spectra were obtained on a luminescence spectrometer (RF5301PC, Shimadzu, Kyoto, Japan) with an external 980 nm laser diode (0–1 W adjustable, continuous wave) as the excitation source. 

### 2.2. Isolation and Identification of Rat CSPCs

#### 2.2.1. Isolation of Rat CSPCs

All procedures involving animals were approved by the Ethics Committee of the Shanghai Jiao Tong University School of Medicine. 

Articular CSPCs were harvested from the knee joints of newborn Sprague Dawley (SD) rats (Sippr-BK Laboratory Animal Co. Ltd., Shanghai, China) via the classic fibronectin assay, as previously described in Refs. [4,27]. Briefly, the neonatal hyaline distal femoral cartilage was incised, washed in sterile chloromycetin and phosphate-buffered saline (PBS), and digested in high-glucose Dulbecco’s modified Eagle’s medium (H-DMEM) containing 0.2% collagenase NB4 (Serva, Heidelberg, Germany) at 37 °C for 6–8 h. The collected digested suspension was filtered using a 200 μL strainer, centrifuged at 1500 rpm for 5 min, resuspended in low-glucose DMEM (L-DMEM), and seeded on fibronectin-treated 100 mm plastic Petri dishes at a density of 1–2 × 10^5^ cells/cm^2^. After incubation at 37 °C for 20 min, the nonadherent cells were discarded with the supernatant, and fresh L-DMEM containing 10% fetal bovine serum (FBS) was added. The harvested cells were cultured at 5% CO_2_ at 37 °C and were not subcultured until an 80% confluence. The CSPCs at passage 3 were used for cell identification and nanoparticle labeling. 

#### 2.2.2. Identification of Rat CSPCs 

For stem cell identification, the expression of CD29 (integrin β1), CD34, CD44, CD45, and CD90 was tested via flow cytometry analysis using fluorescence-conjugated rabbit anti-human antibodies. The results were processed using FlowJo X software (Tree Star, Inc., Ashland, OR, USA). Multilineage-induced differentiation assays were also performed. For adipogenesis and osteogenesis, CSPCs were seeded onto 6-well plates at a density of 1 × 10^6^ cells/well, and adipogenic and osteogenic differentiation media (Cyagen Biosciences Inc., Suzhou, China) were then added. For chondrogenesis, 20 μL of 1 × 10^7^ cells/mL CSPC suspension was placed in the middle of each well of 12-well plates; it was incubated at 5% CO_2_ at 37 °C for 2 h and then added to a chondrogenic differentiation medium (Cyagen Biosciences Inc., Suzhou, China). All media were changed every 2–3 days. The CSPCs for adipogenesis were fixed with 4% paraformaldehyde and dyed with Oil Red O (Sigma-Aldrich Co., St Louis, MO, USA) on the 7th day of induction; the CSPCs for osteogenesis were fixed and dyed with Alizarin red S (Sigma-Aldrich Co., St Louis, MO, USA) on the 10th day of induction; the CSPC pellets for chondrogenesis were fixed and stained with Alcian blue (Sigma-Aldrich Co., St Louis, MO, USA) on the 21st day of induction. 

A real-time polymerase chain reaction (RT-PCR) was also performed to analyze the differentiation capability of CSPCs. On the 7th and 10th days of adipogenesis, 10th and 14th day of osteogenesis, and 21st day of chondrogenesis, the total RNA from the rat CSPCs in each differentiation group was extracted with a TRIzol Reagent (Invitrogen, Carlsbad, CA, USA), and the cDNA was obtained by reverse transcription (RT) according to the manufacturer’s instructions. Rat β-actin was used as an internal control. Data were analyzed using the comparison Ct (2^−^^△△Ct^) method and expressed as a fold change compared with the control. Each sample was analyzed in triplicate. The primer sequences used were as follows: β-actin: forward, 5′-CCTCTATGCCAACACAGT-3′; reverse, 5′-AGCCAC CAATCCACACAG-3′; adiponectin: forward, 5′-CCCGAGAATCAAAGAACAG-3′; reverse, 5′-AACACTCAGAACCCTCAAAGTA-3′; fatty acid-binding protein 2 (AP2): forward, 5′-ATGAAATCACCCCAGATGAC-3′; reverse, 5′-TGCCCTTTNGTAAACTCTTGTA-3′; fatty acid synthase (FAS): forward, 5′-TTGATGAAGAGGGACCATAAAG-3′; reverse, 5′-CAAGGCATTAGG GTTGATGT-3′; osteocalcin: forward, 5′-GAACAGACAAGTCCCACACAG-3′; reverse, 5′-CAGGTCAGAGAGGCAGAATG-3′; RUNX: forward, 5′-CGAAATGCCTCTGCTGTTAT-3′; reverse, 5′-CGTTATGGTCAAAGTGAAACTCT-3′; alkaline phosphatase (ALP): forward, 5′-GAAAGAGAAAGACCCCAGTTAC-3′; reverse, 5′-ATACCATCTCCCAGGAACAT-3′; aggrecan: forward, 5′-ATCTATCGGTGTGAAGTGATG-3′; reverse, 5′-CTCGGTCAAAGTCCAGTGT-3′; collagen II: forward, 5′-GGCGAGTCTTGCGTCTAC-3′; reverse, 5′-GTGCTTCTTCTCCTTGCTCTT-3′; SOX9: forward, 5′-CTTGGCTCCTTCAGAGTTAGT-3′; reverse, 5′-AATCCCCTCAAAATGGTAAT-3′.

### 2.3. Cell Labeling and Cellular Uptake of Nanoparticles

#### 2.3.1. Cell Labeling Procedure

The CSPCs were labeled with the nanoparticles via co-culture, as previously described in Ref. [28]. In brief, after the seeded CSPCs adhered to the dishes, the initial medium containing 10% FBS was replaced with L-DMEM, which contained different concentrations of nanoparticles. The dishes were then placed in the incubator for several hours to achieve internalization; this was followed by the untaken nanoparticles being washed off with PBS.

#### 2.3.2. Quantification of the Nanoparticles Taken by the CSPCs 

To assess the cell labeling efficiency of UCNP@SiO_2_-TS and UCNP@SiO_2_, CSPCs labeled with various co-culture concentrations (50–400 μg/cm^2^) of nanoparticles were collected and analyzed using inductively coupled plasma–atomic emission spectrometry (ICP–AES, iCAP 6000 Radial, Thermo Fisher Scientific, Waltham, MA, USA) to measure the amount of Y^3+^ ions in the cells. To determine the cellular uptake of UCNP@SiO_2_-TS, the labeled CSPCs were also observed and imaged using TEM (H-7650, Hitachi, Tokyo, Japan) at 80 kV. 

#### 2.3.3. Cellular Uptake Observed under UCL

For UCL observation, CSPCs were seeded on 35 mm coverglass-bottom confocal dishes at a density of approximately 2 × 10^4^ cells/well and then co-cultured with different concentrations (20–400 μg/cm^2^) of UCNP@SiO_2_-TS in L-DMEM at 5% CO_2_ at a temperature of 37 °C for 6 h. Subsequently, the cells were fixed with 4% paraformaldehyde, stained with 40,6-diamidino-2-phenylindole (DAPI), and visualized in three channels: bright field, 405 nm laser for DAPI, and NIR excitation for UCL, using two-photon confocal microscopy (TCS SP8 STED 3X, Leica, Wetzlar, Germany) equipped with a continuous-wave (CW) 980 nm laser (2 W/cm^2^). The emission signals were measured at wavelengths of 461 ± 20 nm for DAPI and 470 ± 20 nm for UCL. To discriminate these two emission signals of similar wavelengths, the images of the UCL channel were processed into a green pseudo-color. 

### 2.4. Influence of UCNP@SiO_2_-TS on the Functions of CSPCs

#### 2.4.1. Cell Viability

To investigate the optimal co-culture labeling concentration and time of UCNP@SiO_2_-TS, CSPCs seeded on 96-well plates at a density of 8000/well were co-cultured with ascending concentrations (0–600 μg/cm^2^) of UCNP@SiO_2_-TS in FBS-free L-DMEM for 2, 6, and 12 h. They were then tested using Cell Counting Kit-8 (CCK-8, Dojindo, Japan). Unlabeled cells in L-DMEM containing 10% FBS were used as the control group. The cytotoxicity of Transfectin 3000 was examined by comparing the viability of CSPCs co-cultured with UCNP@SiO_2_-TS and UCNP@SiO_2_ for 6 h. The optical density (OD) was measured at a wavelength of 450 nm. To avoid possible optical interference from the opaque nanoparticles swallowed by the cells, we also measured the OD of the labeled cells with a CCK-8 reagent before reagent incubation, and the results are denoted as OD_UCNP_. Cell viability (%) was calculated as follows: [(OD_test_ − OD_UCNP_)/OD control] × 100%. 

#### 2.4.2. Cell Migration

To evaluate the migration of CSPCs labeled with UCNP@SiO_2_-TS, a wound healing experiment was performed. CSPCs were seeded on 6-well plates at a density of 1 × 10^6^ cells/well and co-cultured with different concentrations (0–100 μg/cm^2^) of UCNP@SiO_2_-TS for 6 h. This was followed by washing away untaken nanoparticles and adding fresh L-DMEM with 10% FBS. Cells incubated in L-DMEM containing 10% FBS without labeling were assayed as a control group. A linear wound in the middle of each well was made using a sterile 100 µL pipette tip. At 0, 6, and 24 h after scratching, images of the wounds were captured from six randomly selected regions under microscopy, and the wound areas were measured using Image-Pro Plus 6.0 software. Wound healing (%) was calculated as follows: [(Area initial − Area test)/Area initial] × 100%.

#### 2.4.3. Multilineage Differentiation 

Multilineage differentiation of the CSPCs labeled with different co-culture concentrations (0–100 μg/cm^2^) of UCNP@SiO_2_-TS was conducted by differentiation staining and PCR assays, as described above for the identification of CSPCs.

### 2.5. Preparation of UCNP@SiO_2_-TS-Labeled CSPCs Encapsulated in Alginate Hydrogel

CSPCs were seeded on 10 cm dishes and labeled with 20 μg/cm^2^ of UCNP@SiO_2_-TS when the cell confluence reached 80%. Subsequently, the labeled CSPCs were encapsulated in alginate hydrogel, as previously described in Ref. [29]. Briefly, sterilized alginate (Sigma-Aldrich Co., St Louis, MO, USA) was dissolved at 2% in deionized water by stirring until the solution became viscous. The labeled CSPCs were then digested and resuspended in 2% alginate solution, diluted, and subdivided into groups of different relative cell concentrations (90, 45, 18, 9, and 4.5 × 10^6^ cells/mL alginate). The corresponding nanoparticle concentrations were 10, 5, 2, 1, and 0.5 mg/mL, respectively. The cell–alginate mixture was then dropped through pipettes into 11.1 g/L of CaCl_2_ solution to turn into hydrogel spheres. After 10 min, the five groups of cell–hydrogel constructs were collected, rinsed with PBS to remove extra CaCl_2_ solution, and incubated in L-DMEM containing 10% FBS for further use. 

### 2.6. In Vitro Tri-Modal Imaging of UCNP@SiO_2_-TS and UCNP@SiO_2_-TS-Labeled CSPCs

#### 2.6.1. In Vitro UCL Imaging 

The cell–hydrogel constructs containing different concentrations (4.5–90 × 10^6^ cells/mL) of UCNP@SiO2-TS-labeled CSPCs were fixed with 4% paraformaldehyde, sectioned into slides, stained with DAPI, and visualized and photographed under a confocal microscope, similar to the method described in Section 2.3.1.

#### 2.6.2. In Vitro CT Imaging

The cell–hydrogel constructs of different concentrations (4.5–90 × 10^6^ cells/mL) were fixed with 4% paraformaldehyde. Corresponding concentrations (0.5–10 mg/mL) of nanoparticles dispersed in alginate hydrogel spheres and suspended in agarose gel were also fabricated to evaluate the influence of different scaffolds and the existence of cells on imaging results. All the nanoparticle–alginate hydrogel spheres (with and without CSPCs) and the nanoparticle–agarose suspension were immobilized at the bottom of 1.5 mL Eppendorf tubes and scanned using a clinical CT scanner (Brilliance 64 CT Scanner, Philips, the Netherlands). The signal intensity (SI) of the 0.02–0.04 cm^2^ regions of interest (ROIs) was measured from the DICOM images using Sante DICOM Viewer Pro Software (Santesoft LTD., Nicosia, Cyprus). Deionized water, L-DMEM, agarose, and alginate hydrogel spheres containing no cells (0 × 10^6^ cells/mL) were also scanned and measured as controls under the following conditions: 80 kV, 400 μA; exposure time of 800 ms; field of view of 50 × 70 mm. The Hounsfield units (HU) were calculated as follows: HU = SI_test_ − SI_water_.

#### 2.6.3. In Vitro MRI Measurement

The same specimens used for the CT imaging were also scanned using MRI. The T1-weighted and T2-weighted spin-echo images were acquired on a 7.05 T scanner (BioSpin MRI GmbH, Biospec 70/20 USR, Bruker, Rheinstetten, Germany). The parameters for the T1 measurement were: echo time (TE) = 6 ms; repetition time (TR) ranging from 110 to 3000 ms. The parameters for the T2 measurement were: TR = 3000 ms; TE ranging from 8.1 to 194.4 ms. The remaining parameters were: flip angle = 90°; matrix = 256 × 256; field of view (FOV) = 70 × 40 mm; slice thickness = 1.0 mm. The longitudinal relaxation time (T1) and transverse relaxation time (T2) values were calculated using the equation: SI = A × e^−TE/T2^ × (1 − e^−TR/T1^), where A is a constant [30,31], and the signal intensity (SI) of the 0.1 cm^2^ ROI was measured from the DICOM images of the corresponding TE or TR using Sante DICOM Viewer Pro Software. The relaxivity coefficients, calculated as the gradient of the plot of r1 (=1/T1) and r2 (=1/T2) versus the molarity of Gd^3+^, were obtained from regression fitting using MATLAB 2016b software (The MathWorks, Inc., Natick, MA, USA). 

### 2.7. Long-Term In Vivo Tri-Modal Tracking of Labeled CSPCs during Ectopic Chondrogenesis

Male nude mice (6–8 weeks, 20–22 g) (Sippr-BK Laboratory Animal Co. Ltd., Shanghai, China) were anesthetized by the inhalation of 2% isoflurane, and the hydrogel constructs containing UCNP@SiO_2_-TS-labeled CSPCs were implanted subcutaneously on the backs of the nude mice. On days 0, 14, and 28 since the implantation, the mice were anesthetized and imaged using the IVIS Lumina III In Vivo Imaging System (PerkinElmer Inc., Waltham, MA, USA), which was equipped with a 980 nm laser as the excitation source; the laser power density was 0.5 W/cm^2^. The UCL signal was collected through an emission filter at 800 ± 10 nm, and the signal intensity of the ROIs was measured. The mice that were engrafted with the constructs were also scanned using CT on days 0, 14, and 28; the parameters were the same as for the in vitro examinations mentioned in Section 2.6.2. The opening of the knee cavity was performed to expose the patellar groove of the femur. We then created osteochondral defects with a 30-gauge needle. The constructs were injected into the defects, and the femoral condyles of mice were dissected en bloc for MRI on days 0, 14, and 28; the parameters were the same as for the in vitro examinations mentioned in Section 2.6.3.

### 2.8. Statistical Analysis

All experiments were repeated at least three times to ensure data validity. The numerical data were analyzed using GraphPad Prism 8 Software (GraphPad Software, San Diego, CA, USA) and presented as the mean ± standard deviation. A one-way ANOVA was employed to evaluate significant differences among groups, and a *t*-test was used to compare two groups. Differences were considered significant at the following levels: * *p* < 0.05, ** *p* < 0.01, *** *p* < 0.001.

## 3. Results

### 3.1. Characterization of UCNPs

The core–shell UCNPs were synthesized using Yb and Tm co-doped NaYF_4_ for the core and undoped NaGdF_4_ for the shell following a previously reported hot-injection approach. Figure 2A shows transmission electron microscopy (TEM) images of the synthesized NaYF_4_:Yb. The UCNPs exhibited a hexagonal plate shape with a diameter of approximately 60 nm. The UCL spectra of UCNPs showed two typical UCL peaks at approximately 470 and 800 nm when excited by a 980 nm laser (Figure 2B).

### 3.2. Identification of Rat CSPCs

As shown in Figure 3A, the CSPCs at passage 3 generally appeared elongated and spindle-shaped in the whirlpool distribution. On the 7th day of adipocytic differentiation, lipid-rich vacuoles stained with Oil red O were observed within the cytoplasm. On the 10th day of osteogenic differentiation, nodules that were positive for Alizarin Red were formed. On the 21st day of chondrogenic differentiation, the CSPCs were all positive for Alcian blue.

To identify whether the isolated cell population was mesenchymal, cell surface markers were analyzed using flow cytometry. The classic positive markers for BMSCs, such as CD29, CD44, and CD90, and negative markers for BMSCs, such as CD34 and CD45, were analyzed. As displayed in Figure 3B, a high expression of positive markers was observed (CD29: 89.9% ± 5.6%, CD44: 24.5% ± 2.2%, CD90: 80.2% ± 7.8%), and a significantly low expression of negative markers was measured.

The RT-PCR results in Figure 3C exhibited significant time-dependent increases in the mRNA expression of adipogenic markers (adiponectin, AP2, and FAS), osteogenic markers (osteocalcin and RUNX2), and chondrogenic markers (aggrecan, collagen II, and SOX-9).

### 3.3. Cell Labeling and Cellular Uptake of Nanoparticles

As shown in Figure 4A, the nanoparticles presented explicit fluorescence with a dose-dependent intensity, which indicates that the nanoparticles were all located homogeneously in the cytoplasm. Owing to the closeness between the emission ranges of DAPI (461 ± 20 nm) and UCL (470 ± 20 nm), the blue UCL fluorescence was pseudo-colored in green. The gaps between the nuclei and nanoparticles are noteworthy, and no nanoparticles were observed in the nuclei. This indicates that no perinuclear aggregation, which was revealed to be responsible for cell function damage, occurred during the UCNP@SiO_2_-TS labeling. Consistent results were obtained from the TEM images; clusters of nanoparticles were encapsulated in endosomes that were diffusively dispersed within the cytoplasm, and the nuclei and all the organelles had a fine morphology (Figure 3B, left column), thereby implying the favorable cytocompatibility of UCNP@SiO_2_-TS. The images of prominent cell membranes encircling the nanoparticles showed clear evidence of the endocytosis mechanism of UCNP@SiO_2_-TS labeling (Figure 4B, right column). 

The uptake of UCNP@SiO_2_ and UCNP@SiO_2_-TS at various co-culture concentrations (50–400 μg/cm^2^) was calculated from the amount of yttrium measured by ICP–AES. A significant time-dependent increase in the uptake was observed. The uptake curve of UCNP@SiO_2_-TS presented a significantly steadier dose-dependent linear increase compared with UCNP@SiO_2_, which indicates that the labeling efficiency of UCNP@SiO_2_ was improved by the modification using Transfectin 3000 (Figure 4C). The labeling efficiency of UCNP@SiO_2_-TS at 20 μg/cm^2^ for 6 h measured 114.36 ± 8.7 pg/cell, which was obtained from a calculation using the linear slope.

### 3.4. The Cytotoxicity of UCNP@SiO_2_-TS on CSPCs

To determine the optimal co-culture time and concentration for the labeling of UCNP@SiO_2_-TS, CSPCs were incubated with ascending concentrations (0–100 μg/cm^2^) of nanoparticles for three different time periods (6, 12, and 24 h). The OD values of nanoparticle-labeled CSPCs that were added with a CCK-8 reagent before the incubation reaction were measured, and the results exhibited a significant proportional correlation with the nanoparticle concentrations (Figure 5A). Thus, it was necessary to exclude the disturbance of the existence of the nanoparticles in the CCK-8 assay. Consequently, the calculation [(OD_test_ − OD_UCNP_)/OD control] × 100% was applied. To examine the cytotoxicity of Tranfectin 3000, cell viability was compared between modified and unmodified UCNP@SiO_2_ at various concentrations, and no significant difference was observed (Figure 5B), thereby proving the biosafety of Tranfectin 3000. As demonstrated in Figure 4C, the cell viability subsequent to a co-culture with ascending concentrations (0–400 μg/cm^2^) of UCNP@SiO_2_-TS for different time periods (6, 12, and 24 h) was analyzed. It is noteworthy that the cells were co-cultured with nanoparticles suspended in FBS-free DMEM in this study; therefore, both cells incubated in DMEM, with and without FBS, were tested as two types of controls. The 50% lethal dose (LD50) of all three time groups was approximately 500 μg/cm^2^. In the concentration range, from 0 to 250 μg/cm^2^, no significant difference was observed between the 6 and 12 h groups, while the 24 h group presented a significant decrease in viability. As shown in Figure 5D, wound healing was significantly impaired when the concentration was greater than 50 μg/cm^2^. The results of multi-differentiation are displayed in Figure 5E. On the 7th day of adipogenesis, fewer vacuoles stained with Oil red O were observed when the concentration exceeded 50 μg/cm^2^, which was in agreement with the expression of FAS. However, the expressions of AP2 and adiponectin were reduced at a concentration of 20 μg/cm^2^. Surprisingly, on the 10th day of osteogenesis, the positive staining of Alizarin Red visually increased with a higher concentration of nanoparticles. On the contrary, the expression of osteogenic markers (osteocalcin, RUNX2, and ALP) showed a significant decreasing trend when the concentration exceeded 50 μg/cm^2^. After 2 weeks of chondrogenic induction, each group was stained positive for Alcian blue, while 20 μg/cm^2^ displayed stronger staining than the other groups did. The expression of the cartilage-related genes, aggrecan, SOX9, and collagen II, was upregulated in 20 μg/cm^2^. According to the results above, we applied 20 μg/cm^2^ for 6 h as a preferable co-culture condition for UCNP@SiO_2_-TS.

### 3.5. In Vitro and In Vivo CT Imaging of CSPCs

To evaluate the CT features of UCNP@SiO_2_-TS, in vitro phantom CT imaging of UCNP@SiO_2_-TS at increasing concentrations (0.5–10 mg/mL) and hydrogel constructs containing UCNP@SiO_2_-TS-labeled CSPCs of increasing numbers (4.5–90 × 10^6^ cells/mL) were performed. A significant proportional increase in signal intensity was observed in the CT images and CT value profile (Figure 6A) in accordance with the increasing opacity of the constructs (Figure 5C). No significant differences were observed within the UCNP@SiO_2_-TS suspended in agarose, UCNP@SiO_2_-TS encapsulated in alginate hydrogel, and hydrogel constructs containing UCNP@SiO_2_-TS-labeled CSPCs, which contained identical relative concentrations of nanoparticles (Figure 6B). This result suggests that the existence of different scaffolds or cells does not interfere with CT imaging. The in vivo CT images obtained on days 0, 14, and 28 post-implantation presented consistent signals of the implanted constructs, and it was difficult to discern the constructs from ambient tissue in groups of fewer than 18 × 10^6^ cells/mL (2 mg/mL of nanoparticles) (Figure 7). 

### 3.6. In Vitro and In Vivo MRI Imaging of CSPCs

Figure 8A displays the representative T1-weighted (TR = 800 ms, TE = 6 ms) and T2-weighted (TR = 3000 ms, TE = 8.1 ms) MR phantom images of UCNP@SiO_2_-TS-labeled CSPCs of various concentrations. It is noteworthy that the signal intensities in all the T1-weighted images tended to decline with Gd^3+^ concentration, and the signals were minimal when the nanoparticle concentrations exceeded 5 mg/mL (which corresponds to a 4.99 mol/L Gd^3+^ concentration), which was inconsistent with the widely evaluated positive MRI contrast capabilities of gadolinium [16,32,33,34]. A similar signal intensity attenuation was also observed at a manganese concentration of 1.5 mM in a previous study [35]. It is worth noting that the Gd^3+^ concentrations examined in this study were no lower than 1.25 mmol/L (mM), whereas the Gd^3+^ concentrations tested for MRI in most studies were significantly lower. According to the linear fitting curves, the longitudinal relaxivity (r1) and transverse relaxivity (r2) were measured at 1.182 and 10.83 mM^−1^S^−1^, respectively (Figure 8B), which were lower than those calculated in previous studies that applied the same type of UCNPs [11,14,15,17,36].

MRI T1-weighted contrasts can also be used in vivo to image the UCNPs-labeled CSPCs in a cartilage-defect model at days 0, 14, and 28 (Figure 9). The constructs of UCNPs-labeled CSPCs on MRI images were marked in red arrows. However, the quality of the MRI images was poor.

### 3.7. In Vitro and In Vivo UCL Imaging of Labeled CSPCs in Hydrogel Constructs

As shown in Figure 10, the nanoparticles in the hydrogel constructs presented explicit fluorescence with a dose-dependent intensity, which indicates that the nanoparticles were all located homogeneously in the cytoplasm. The result was similar to that shown in Section 3.3.

Long-term tracking and quantification of CSPCs engrafted with UCL was also performed on days 0, 14, and 28 post-implantation. As displayed in Figure 11A, the intensity of the UCL collected at 800 nm increased with the number of gradient cells. Although the luminescence gradually attenuated over time, the signals remained explicit on day 28, even in the group with the lowest cell number. Proportional results were consistently obtained from the plots of the quantified UCL signal intensity versus the tracked cell numbers. Among the groups with cell numbers ranging from 4.5 to 45 × 10^6^ cells/mL, no significant difference was observed between days 14 and 28, and both were slightly lower than that on day 0. However, the intensity of the group of 90 × 10^6^ cells/mL on day 28 was remarkably reduced, which we consider to be a result of the expansion of the labeled cells. 

## 4. Discussion

With the advancement of nanoplatforms in molecular imaging, UCNP remains an active area of research in materials or clinical translation. Numerous innovative achievements have been reported regarding tumor-targeted imaging and photodynamic therapy with tri-modality, including UCL, CT, and MRI, thereby indicating the tremendous translational potential of the tri-modal nanoplatform based on UCNPs. From our perspective, such a platform is also of great value for the translation of implanted stem cell tracking. Therefore, we exploited a synergetic UCNP-based tri-modal nanoplatform for the tracking of CSPCs during ectopic chondrogenesis. 

In agreement with the existing research, the labeling procedure of UCNPs is quite simple, i.e., the nanoparticles can be swallowed by cells through mere co-culture for several hours by endocytosis at 37 °C. Liu et al. proved that the cellular uptake of UCNPs was energy-dependent, because UCNP uptake was completely inhibited at 4 °C [37]. In our study, the TEM images in Figure 4B also provided visual evidence for the mechanism of endocytosis, and the intactness of all the organelles and nuclei indicated the biocompatibility of UCNP. With the aid of Transfectin 3000, which is a suspension of positively charged nano-polymers that are commercially used for transfection through endocytosis, the labeling efficiency and stability were significantly improved (Figure 4C), which further proves the endocytosis mechanism. However, extrusion and compression cannot be avoided with an overdose of UCNPs, owing to the limited space in the cytoplasm, and excess amounts of foreign bodies, such as UCNPs, would still damage the cell functions. Therefore, the proper labeling concentration and time should be determined. We have noted that the co-culture concentrations and time lengths of UCNPs for stem cells in various studies were remarkably different; concentrations ranged from 25 to 1000 µg/mL and times ranged from 2 to 24 h [11,25,38,39,40,41,42]. Considering that the concentrations denoted by µg/mL may be misleading, we applied µg/cm^2^ as the unit of co-culture concentration throughout this study. As shown in Figure 5E, the multi-lineage differentiation capabilities were significantly impaired when the co-culture was higher than 20 µg/cm^2^ (approximately 100 µg/mL in a 10 cm dish), which is similar to the results obtained in other studies [28,41]. Taking the UCL image results in Figure 4A and time-dependent uptake efficiency presented in Figure 4C into consideration, we finally applied 20 µg/cm^2^ and 6 h as the co-culture concentration and time, respectively.

Because the tracking of implanted cells typically requires retention of the markers in the cells for as long as possible, extrusion of the nanoparticles from the cells needs to be avoided. Several Transwell assays have shown that ligand-modified UCNPs can be retained in labeled cells for as long as 14 days [21,37,38]. In this study, the CT signals and most UCL signals remained stable with no significant attenuation over 28 days (Figure 6 and Figure 11). The observation time was limited to 28 days because the animals were euthanized on day 28, leaving the signal follow-up undetected. However, the possibility of the nanoparticles being extruded by the labeled CSPCs and internalized by nearby cells, such as phagocytes, cannot be excluded by the long-term explicit signals of multimodal imaging [43]. Abraham et al. observed that most of the ferumoxides used as cell labelers were taken by macrophages released from nonsurviving cells on the 21st day post-implantation [5]. Thus, the survival of the implanted cells may also be estimated by the multimodal imaging of UCNPs. The MRI results were unexpected. Gadolinium has been proven to be T1 paramagnetic, i.e., the signal intensity increases with Gd concentration because the longitudinal relaxation time can be shortened by Gd. However, the opposite was observed in this study because the concentrations we studied were significantly higher than those of other studies on the MRI features of Gd-doped UCNPs. Therefore, there are still limitations in the MRI properties of UCNP.

We have to admit that the UCNPs-based stem cell tracking platform we studied is still far from satisfactory. Although the lanthanide-doped nanoparticles are generally considered as chemically nontoxic, little has been learnt about the interaction between UCNPs and the labeled cells. A standardized protocol for the assessment of cytotoxicity should be made. Apart from the concentration, composition, and exposure time, more interdependent parameters should be taken into account [43]. The UCNPs used as a multi-modal bioprobe are also in demand of tremendous modification. First, the intake of nanoparticles was relatively limited by the size, and the cellular damage caused by the nanoparticles as foreign bodies would lead to a shorter time of retention in the cell. Therefore, nanoparticles of smaller size are desired. It has been proven that sub-10 nm nanoparticles could be obtained through replacing NaYF4, the regularly used host material, with NaLuF4, and enhanced UCL could be obtained [9]. In addition, bioprobes emitting light over 1000 nm, such as Ho^3+^- or Er^3+^-doped nanoparticles, and YAG:Nd^3+^ nanocrystals, would bring about lower background autofluorescence [44]. The UCNPs we employed are limited by the broad spectral fingerprint, and a narrower emission range is expected. 

## 5. Conclusions

In this study, a platform for in vivo upconversion fluorescence imaging of UCL, MRI, and CT multi-mode animal living cells was constructed based on the upconversion characteristics of upconversion nanomaterial and the characteristics of both MRI and CT imaging. In the classic hypodermic ectopic chondrogenesis model of nude mice, the high sensitivity of UCL and the accurate spatial resolution of CT imaging were fully developed, and the location and relative quantification of UCNP-labeled chondrocytes in a basic ectopic chondrogenesis period (28 days) were realized.

The in vivo tracer method used after stem cell transplantation is not perfect. Therefore, investigating the outcome of the multimodal cell tracer method is of significance. In terms of the cost of operation and the efficiency of imaging, UCNPs have a clear advantage. Owing to its biological safety, more species cells can be investigated, enabling a wide variety of study. The regeneration model can be utilized for different types of stem cells, and UCNPs can be utilized for specific applications in all areas of development.

## Figures and Tables

**Figure 1 biomolecules-11-00958-f001:**
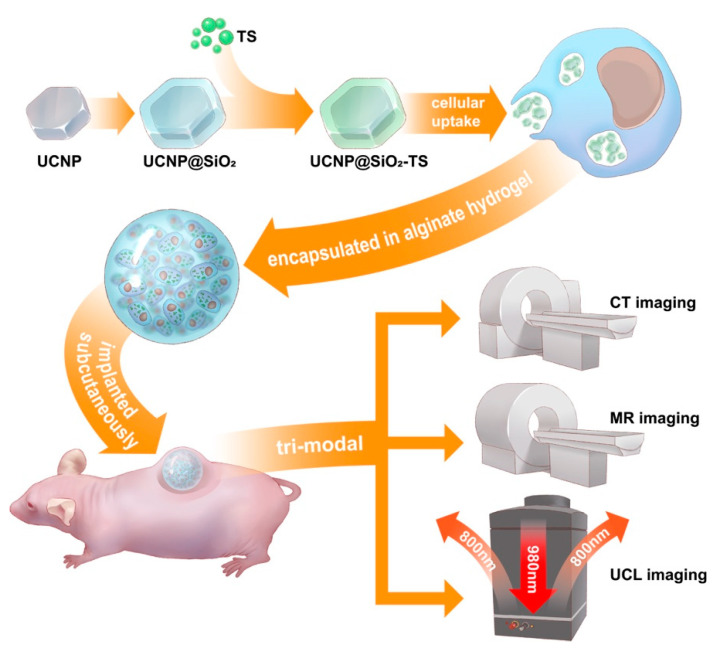
A schematic diagram of the study.

**Figure 2 biomolecules-11-00958-f002:**
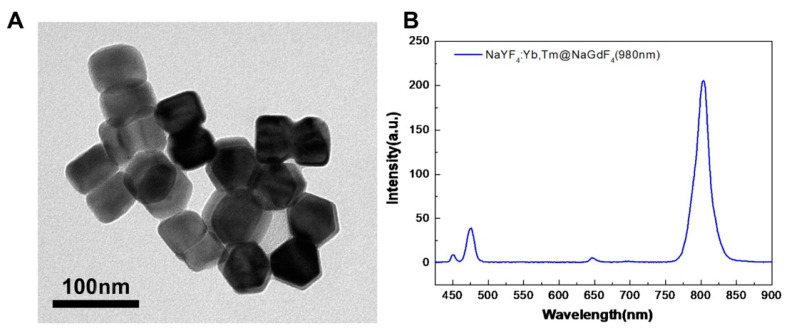
Characterizations of UCNP@SiO_2_-TS (NaYF4:Yb^3+^/Tm^3+^@ NaGdF4 @ SiO_2_-NH2). (**A**) The high-resolution TEM image exhibits a regular hexagonal shape of the particles. (**B**) Luminescence spectrum of UCNP@SiO_2_-TS under 980 nm laser-light excitation. Two peaks can be observed at approximately 470 and 800 nm.

**Figure 3 biomolecules-11-00958-f003:**
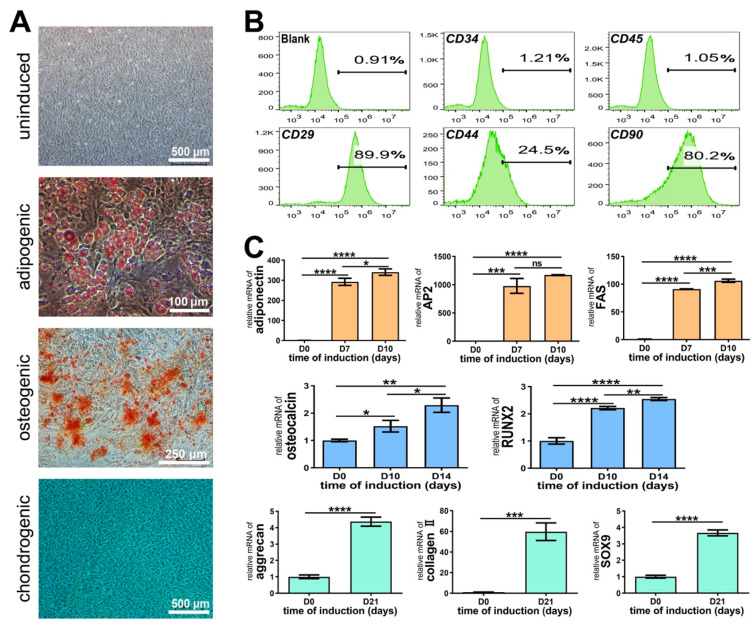
Identification of rat CSPCs. (**A**) Multilineage differentiation of the isolated CSPCs. On the 7th day of adipocytic differentiation, lipid-rich vacuoles stained with Oil red O are observed within the cytoplasm. On the 10th day of osteogenic differentiation, calcium nodules that are positive for Alizarin Red are formed. On the 21st day of chondrogenic differentiation, the CSPCs are all positive for Alcian blue. (**B**) Flow cytometry analysis of the expression of cell-surface markers. The surface markers that are specifically positive for mesenchyme cells (CD29, CD44, and CD90) are all highly expressed, whereas CD34 and CD45 are negatively expressed. (**C**) The RT-PCR results show a significant increase in mRNA expression of adipogenic markers (adiponectin, AP2, and FAS), osteogenic markers (osteocalcin and RUNX2), and chondrogenic markers (aggrecan, collagen II, and SOX-9) after multilineage-induced differentiation (* *p* < 0.05, ** *p* < 0.01, *** *p* < 0.001, **** *p* < 0.0001, ns *p* > 0.05).

**Figure 4 biomolecules-11-00958-f004:**
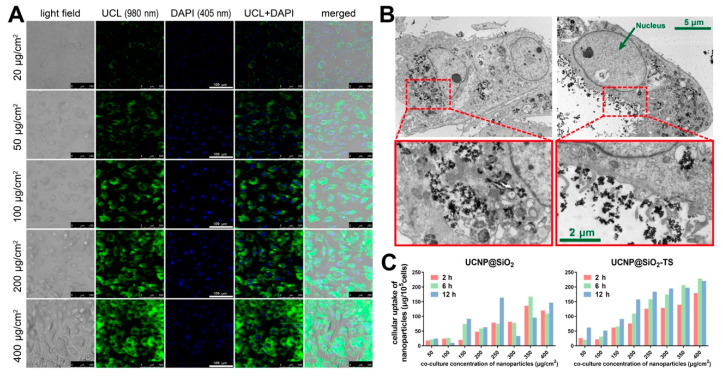
Intracellular uptake of the nanoparticles. (**A**) UCL images of CSPCs after 6 h of co-culture with ascending concentrations (20–400 μg/cm^2^) of UCNP@SiO_2_-TS under confocal microscopy with the excitation of a CW 980 nm NIR laser; scalebar: 100 μm. The cell nuclei are counterstained with DAPI (blue fluorescence). The UCNP@SiO_2_-TS emits strong fluorescence (collected at 470 ± 20 nm) (visually blue, pseudo-colored in green) homogeneously in the cytoplasm, illustrating the spatial relation between the cells and nanoparticles internalized. All images are obtained under the same condition and captured at an identical intensity scale. (**B**) Upper row: Typical TEM images of CSPCs labeled with 50 μg/cm^2^ of UCNP@SiO_2_-TS; scalebar: 5 μm. Lower row: magnified features of vesicles encapsulating nanoparticles diffusive as tiny dark clusters in cytoplasm (left panel) and representing a prominent membrane encircling the nanoparticle clusters during endocytosis (right panel); scalebar: 2 μm. The cell nuclei are all intact and no nanoparticles are found inside. (**C**) Cellular uptake profiles of UCNP@SiO_2_ and UCNP@SiO_2_-TS at various co-culture concentrations (50–400 μg/cm^2^) calculated from the amount of yttrium measured by ICP–AES. A significant time-dependent increase of the uptake is observed. The uptake curve of UCNP@SiO_2_-TS presents a significantly steadier dose-dependent linear increase than UCNP@SiO_2_, indicating that the labeling efficiency of the nanoparticles is improved by the modification using Transfectin 3000. The labeling efficiency of UCNP@SiO_2_-TS at 20 μg/cm^2^ for 6 h measures 114.36 ± 8.7 pg/cell, which is calculated via the linear slope.

**Figure 5 biomolecules-11-00958-f005:**
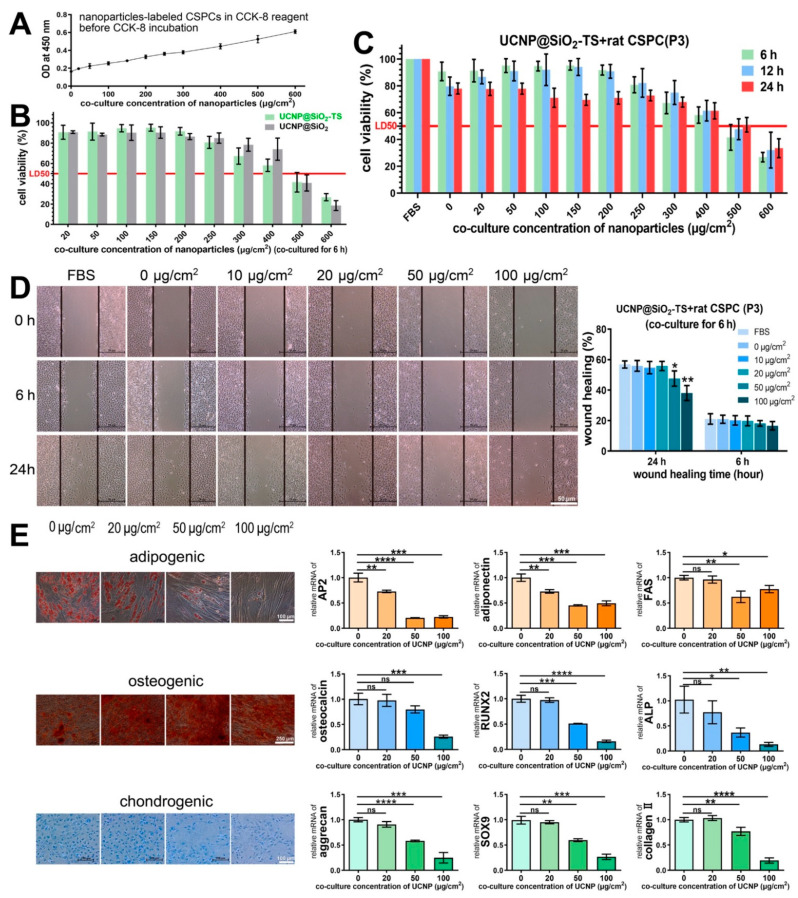
Influence of UCNP@SiO_2_-TS on the functions of CSPCs. (**A**) OD values of nanoparticle-labeled CSPCs added with the CCK-8 reagent before the incubation reaction. The OD values display significant proportional correlations to the concentration of nanoparticles (*p* < 0.05, n = 6), thereby making it necessary to exclude the disturbance caused by the existence of the nanoparticles when testing with CCK-8. (**B**) There is no significant difference between the groups with and without modification using Transfectin 3000 (*p* > 0.05, n = 6), thereby indicating its biosafety. (**C**) Cell viability after co-culture with ascending concentrations (0–400 μg/cm^2^) of UCNP@SiO_2_-TS for different time courses (6, 12, and 24 h). The 50% lethal dose (LD50) of all three time groups is approximately 500 μg/cm^2^. In the concentration range from 0 to 250 μg/cm^2^, no significant difference is observed between the 6 and 12 h group; however, the 24 h group presents a significant decrease in viability (*p* < 0.05, n = 6). **(D**) Wound healing results of CSPCs incubated with varied concentrations (0–100 μg/cm^2^) for 6 h. Wound healing is significantly impaired when the concentration is over 50 μg/cm^2^ (*p* < 0.05, n = 6). (**E**) Multi-differentiation of labeled CSPCs and unlabeled CSPCs. On the 7th day of adipogenesis, fewer vacuoles stained with Oil red O are observed when the concentration reaches over 50 μg/cm^2^, which is in agreement with the expression of FAS, whereas the expression of AP2 and adiponectin begin to reduce at a concentration of 20 μg/cm^2^ (*p* < 0.05, n = 3). On the 10th day of osteogenesis, positive staining of Alizarin Red visually increases with rising nanoparticle concentration. However, the expressions of osteogenic markers (osteocalcin, RUNX2, ALP) exhibit a significant decrease when the concentration is over 50 μg/cm^2^. After 2 weeks of chondrogenic induction, each group was stained positive for Alcian blue, while 20 μg/cm^2^ displayed stronger staining than the other groups did. The expression of the cartilage-related genes, aggrecan, SOX9, and collagen II, was upregulated in 20 μg/cm^2^. (*p* < 0.05, n = 3) (* *p* < 0.05, ** *p* < 0.01, *** *p* < 0.001, **** *p* < 0.0001, ns *p* > 0.05)).

**Figure 6 biomolecules-11-00958-f006:**
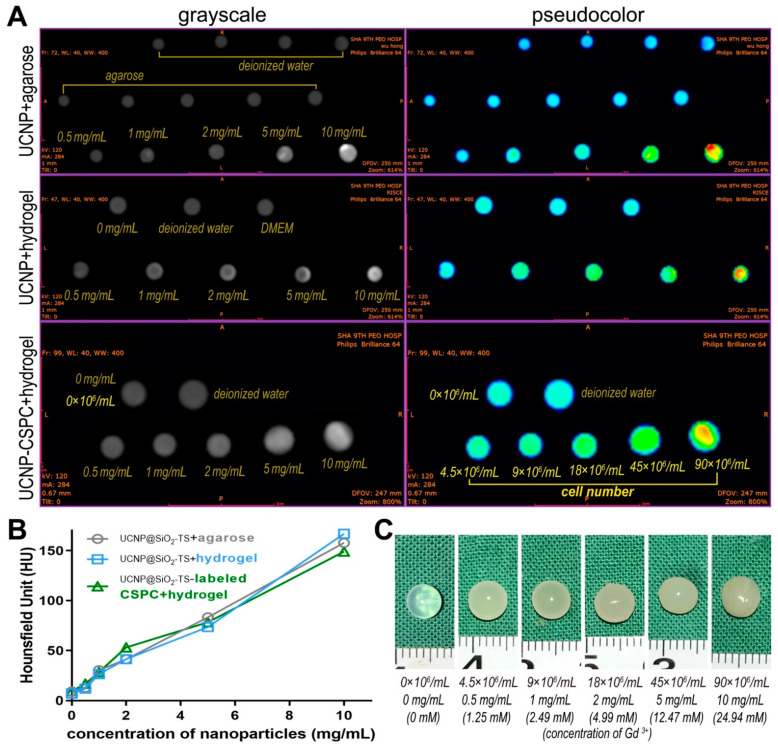
CT imaging of UCNP@SiO_2_-TS of increasing concentrations (0.5–10 mg/mL) and hydrogel constructs containing UCNP@SiO_2_-TS-labeled CSPCs of increasing numbers (4.5–90 × 10^6^ cells/mL). (**A**) CT images of UCNP@SiO_2_-TS suspended in agarose, UCNP@SiO_2_-TS encapsulated in alginate hydrogel, and hydrogel constructs containing UCNP@SiO_2_-TS-labeled CSPCs, which contain identical relative concentrations of nanoparticles. Left column: the original images in grayscale mode; right column: images processed into pseudo-color mode. (**B**) CT values of the three groups. No significant difference exists between the three linear regression results (*p* < 0.05, n = 3). (**C**) Gross view of the hydrogel constructs containing UCNP@SiO_2_-TS-labeled CSPCs examined by CT.

**Figure 7 biomolecules-11-00958-f007:**
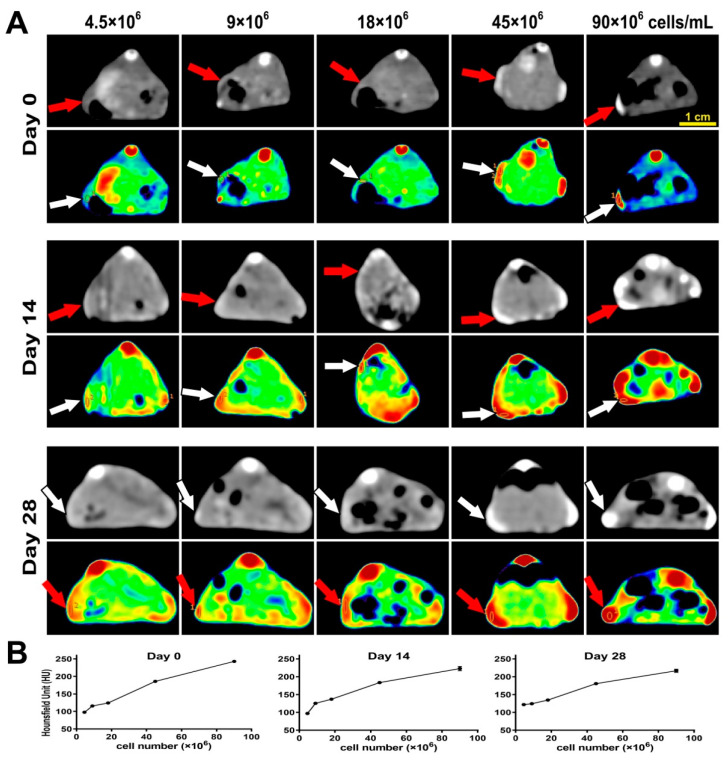
In vivo CT imaging of hydrogel constructs containing UCNP@SiO_2_-TS-labeled CSPCs of increasing numbers (4.5–90 × 10^6^ cells/mL). (**A**) In vivo transverse CT images of the implanted constructs (indicated by red arrows in greyscale and white arrows in pseudo-color) on days 0, 14, and 28 post-implantation. (**B)** Corresponding CT values of the regions of interest (denoted by yellow circles in A).

**Figure 8 biomolecules-11-00958-f008:**
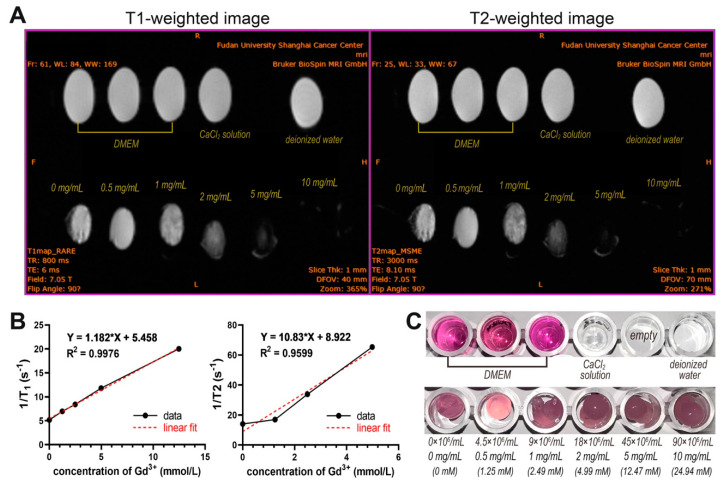
MRI of UCNP@SiO_2_-TS of increasing concentrations (0.5–10 mg/mL) and hydrogel constructs containing UCNP@SiO_2_-TS-labeled CSPCs of increasing numbers (4.5–90 × 10^6^ cells/mL). (**A**) Representative T1-weighted (TR = 800 ms, TE = 6 ms) and T2-weighted (TR = 3000 ms, TE = 8.1 ms) MR phantom images of UCNP@SiO_2_-TS-labeled CSPCs of various concentrations. The signal intensity declines with Gd^3+^ concentration, and the signals can hardly be measured when the nanoparticle concentrations are above 5 mg/mL (which corresponds to 4.99 mol/L of Gd^3+^ concentration), which is inconsistent with the MRI capability of gadolinium. (**B**) According to the linear fitting curve, the longitudinal relaxivity (r1) is measured at 1.182 mM^−1^S^−1^, and the transverse relaxivity (r2) is measured at 10.83 mM^−1^S^−1^. (**C**) Gross view of the MRI phantoms.

**Figure 9 biomolecules-11-00958-f009:**
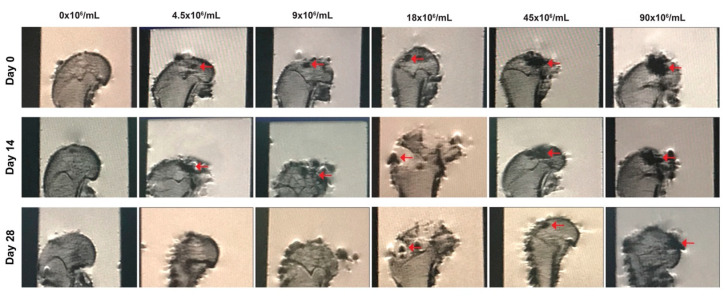
In vivo MRI imaging of hydrogel constructs containing UCNP@SiO_2_-TS-labeled CSPCs of increasing numbers (4.5–90 × 10^6^ cells/mL) in cartilage-defect model. MRI of the injected constructs (indicated by red arrows) on days 0, 14, and 28.

**Figure 10 biomolecules-11-00958-f010:**
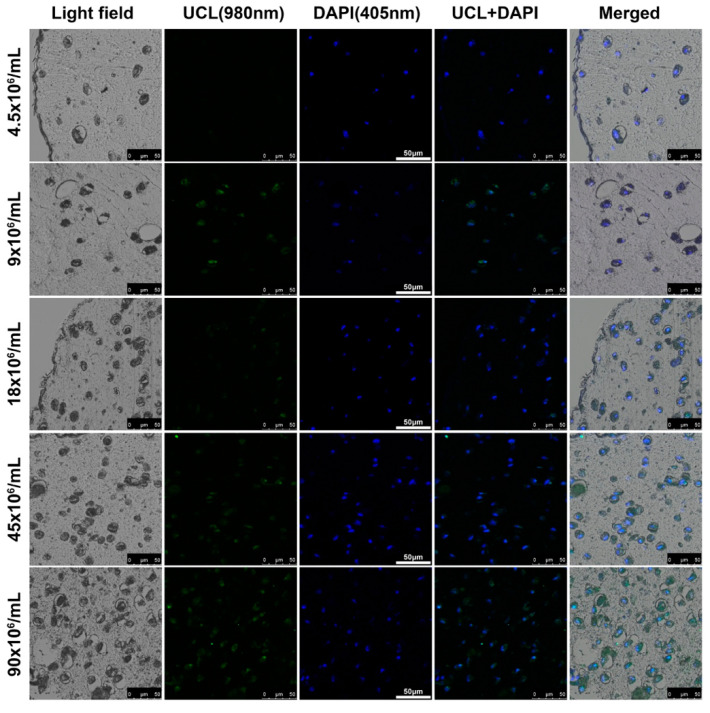
UCL imaging of UCNP@SiO_2_-TS of increasing concentrations (0.5–10 mg/mL) and hydrogel constructs containing UCNP@SiO_2_-TS-labeled CSPCs of increasing numbers (4.5–90 × 10^6^ cells/mL). The cell nuclei are counterstained with DAPI (blue fluorescence). The UCNP@SiO_2_-TS emits fluorescence (visually blue, pseudo-colored in green) homogeneously in the cytoplasm. Scale bar: 100 μm.

**Figure 11 biomolecules-11-00958-f011:**
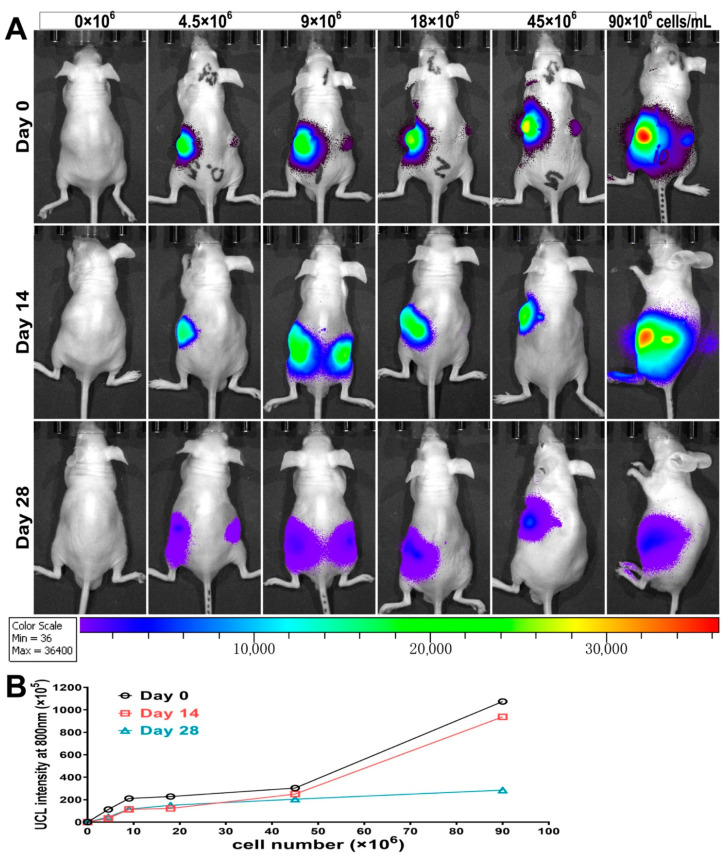
Long-term tracking and quantification of CSPCs engrafted using UCL. (**A**) UCL images of nude rats subcutaneously implanted with hydrogel constructs containing UCNP@SiO_2_-TS-labeled CSPCs of increasing numbers (4.5–90 × 10^6^ cells/mL) taken on days 0, 14, and 28 post-implantation (UCL emission collected at 800 nm in pseudo-color). (**B**) Plots of quantified UCL signal intensity versus the tracked cell numbers.

## Data Availability

Data available on request due to restrictions, e.g., privacy or ethical.

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
