# Peer review of "Long-Term Tri-Modal In Vivo Tracking of Engrafted Cartilage-Derived Stem/Progenitor Cells Based on Upconversion Nanoparticles"

_biomolecules, 2021, doi:10.3390/biom11070958_

Round 1
Reviewer 1 Report
The paper entitled “Long-term Tri-modal in Vivo tracking of Engrafted Cartilage-derived Stem/Progenitor Cells Based on Upconversion Nano-particles” presents interesting results regarding the use of multifunctional nanoparticles for long term (over one month) tracking. Despite a thorough and detailed study, there are some inconsistencies that need to be corrected before publication:
(1) The sentence line 60-61 is not correct and should modify accordingly: “UCNPs are compounds that primarily comprise lanthanide elements from rare earth materials, and they can emit visible near-infrared (NIR) (650–900 450-800 nm) light when evoked by light with a wavelength of 980 nm”. In particular, in this paper, the authors use a Yb,Tm-doped system and its emission around 470 nm.
(2) The authors state that “[UCNPs exhibit] lower autofluorescence and photodamage than the traditional “downconversion” fluorophores”. This statement is true; however, the UCNPs are not ideal for in vivo imaging as the emitting light is situated in the visible range, where biological tissues autofluoresence. The authors should mention that current research is oriented towards the development of new nanoprobes that can be excited and emit in the near-infrared (NIR). They can quote the following papers:
-Exploiting the Biological Windows: Current Perspectives on Fluorescent Bioprobes Emitting above 1000 nm, Eva Hemmer, Antonio Benayas, François Légaré, Fiorenzo Vetrone, Nanoscale Horizons (2012)
- Autofluorescence-free in vivo imaging using polymer-stabilized Nd3+-doped YAG nanocrystals, A. Cantarano, J. Yao, M. Matulionyte, J. Lifante, A. Benayas, D. Ortgies, F. Vetrone, A. Ibanez, C. Gérardin, D. Jaque, G. Dantelle, ACS Applied Materials & Interfaces 12(46) (2020) 51273-51284
(3) The scale of Figure 1 is wrong. I believe, the scale bar represents 100 nm (and not µm as indicated).
(4) The sentence “Figure 1A shows transmission electron microscopy (TEM) images of the synthesized NaYF4:Yb, Tm core and NaYF4:Yb, Tm@NaGdF4 core/shell nanocrystals” is wrong. There is only one image on Figure 1A. Does it represent NaYF4:Yb,Tm core particles or NaYF4:Yb,Tm@NaGdF4 core/shell particles ?
(5) The authors state on line 327 “The core nanoparticles were spherical with a diameter of ~40 nm”. How do the author evidence the shell?
I suggest the authors should present 3 images: the core nanoparticles, the core/shell nanoparticles and the core/shell nanoparticles with the SiO2 layer. It would be a good mean to evaluate the size of the particles and coatings.
(6) On line 369, the authors mention “Owing to the closeness between the emission ranges of DAPI(461 ± 20 nm) and UCL (470 ± 20 nm) …” This statement is not consistent with previous results (line 329), where the authors mention that the UC peak was situated at 461 nm. The authors should review these sentences. It seems that both DAPI and UC peak overlap, which is problematic to discriminate both signals.
(7)The overall bibliography is lacking some essential papers. I suggest the authors add the following references and discuss their results in the light of previous results reported in those papers.
- Recent Advance of Biological Molecular Imaging Based on Lanthanide-Doped Upconversion-Luminescent Nanomaterials, Nanomaterials 4(1):129-154, DOI: 10.3390/nano4010129
- Upconverting nanoparticles: a versatile platform for wide-field two-photon microscopy and multi-modal in vivo imaging. https://doi.org/10.1039/C4CS00173G
- Functionalized Upconversion Nanoparticles: Versatile Nanoplatforms for Translational Research, Curr Mol Med. 2013 Dec; 13(10): 1613–1632.
Reviewer 2 Report
The authors developed platform for tri-modal in-vivo imaging of labeled cartilage-derived stem/progenitor cells. The properties of the developed platform have been assessed in vitro and/or in vivo.
I have some comments/questions:
General remark
A short paragraph and diagram showing the idea behind the developed platform (e.g. structure of the UCNP i.e. core, shell and coatings together with their functionalities would be very helpful). It can be obviously deduced from the manuscript, but such an overview could help the reader to catch the idea.
Figures
Please check the quality of the Figures. There are plenty of subfigures/subdiagrams in the Figures. Especially Figure 5 is completely unreadable (e.g. bitmap quality). Maybe the option is to separate the Figures with high informational content (e.g. Figure 4) into 2 or 3 smaller ones?
Page 2, lines 84-87
The aim of the study also assumed standardization:
“there is insufficient research on standardized imaging features for quantification and real-time in vivo imaging of implanted cells with multi-modality for a long period.”, “Herein, we aim to determine a standard configuration of UCNP labeling … ”
What was the meaning of the proposed standardization? I couldn’t find any signs of standardization as a result of investigation.
Page 2, line 58
„Recently, upconversion nanoparticles (UCNPs) have attracted increased attention …”
What is the meaning of “recently”? It suggests that they are freshly discovered. Please specify.
Please give reference to reasonable review article(s).
Page 2. Line 76
Ao et al. ? According to References it should be Hu et al. Please check.
Page 2, line 77
As in the previous remark – please check.
Page 2, line 77
“… according to a previously developed procedure.”
Please give reference to the work describing the procedure.
Page 5, line 230
“To evaluate the migration of CSPCs labeled with UCNP@SiO2-TS, a wound healing experiment was performed.”
What are possible, real applications of the developed platform? Applications in regenerative medicine?
Page 6, line 285-286 (2.63)
Regarding T1 measurements (mapping) somewhat strange method used (varying repetition time). When working on phantoms (total acquisition time is not a crucial factor), why not to use the most reliable, classical Inversion Recovery method?
Page 6, lines 306-309
Please, give technical details of the CT and MRI including imaging sequences and parameters of in vivo scans. The same as for in vitro studies?
Page 7, line 327, Figure 1
Are NPs really hexagonal plates with diameter of ~60 nm? Spherical core diameter of ~40 nm? Cannot find any … External/internal structure is not shown in Figure 1. Maybe the authors used another data set to assess the structural/geometrical characteristics. If yes, please show the data in the manuscript.
Pages 12-13, lines 491-516 (3.6)
Section title: “3.6 In vitro and in vivo MRI imaging of CSPCs”. But there is no in vivo data presented in this section!
The authors were unable to achieve positive contrast. Maybe the reason is final size of the system (Chem. Mater. 2011, 23, 21, 4877) ? I had the same problem – nanoparticles alone allowed to obtain positive MRI contrast but encapsulated ones not.
Pages 13-14, lines 518-536
Similar situation. The section title is “3.7 In vitro and in vivo UCL imaging of labeled CSPCs in hydrogel constructs” bu there is no in vitro data presented!
Figure 6.
In the Fig. 6A one can read some scan details including imaging sequence names: T1map_RARE and T2_map_MSME. It is inconsistent with 2.6.3 in Materials and Methods. Please improve.
Round 2
Reviewer 1 Report
The authors amended the paper as requested. I suggest to accept the paper for publication.